# Pen-and-Paper versus Computer-Mediated Writing Modality as a New Dimension of Task Complexity

Olena Vasylets [1,*] and Javier Marín [2]

1    Department of Modern Languages, Faculty of Philology and Communication, University of Barcelona, 08007 Barcelona, Spain
2    Departamento de Psicología Básica y Metodología, Facultad de Psicología, Universidad de Murcia, 30100 Murcia, Spain; jms@um.es
*    Correspondence: vasylets@ub.edu

**Abstract:** In this paper we make a proposal that writing modality (pen-and-paper versus computer-based writing can be conceptualized as a cognitive task complexity factor. To lay ground for this theoretical proposal, we first review previous adaptations of cognitive task-based models to second language (L2) writing. We then compare pen-and-paper and computer-based writing modalities in terms of their general characteristics, outline the main tenets of multidisciplinary theoretical models which attribute learning and performance-related importance to writing modality, and review the available empirical evidence. From this we draw theoretical and empirical justification for our conceptualization of writing modality as a task complexity dimension. After outlining our conceptual view, we proceed with the review of the methods which could be used to independently assess cognitive load in paper and computer-written L2 tasks. In the conclusion, implications and suggestions for future research are provided.

**Keywords:** second language writing; pen-and-paper writing; computer-mediated writing; task complexity; task-based language teaching





## 1. Introduction

When Friedrich Nietzsche's vision started to fail, he resorted to the Malling-Hansen writing ball, which was an odd typewriting machine. The interesting fact, though, is that the use of the typing machine affected his writing style, leading Nietzsche to acknowledge that "writing equipment takes part in the forming of our thoughts" (cited in "The Shallows" by N. Carr 2010, p. 21). Today, evidence from neuroscience, educational psychology and cognitive writing research has demonstrated that language learning and performance may differ depending on whether a writer employs pen and paper as opposed to using keyboard and screen (Chan et al. 2017; Ihara et al. 2021). In spite of these findings, the area of L2 writing research has paid little attention to the idiosyncrasy of learning and performance in L2 writing tasks in paper-based versus computer-based writing modalities. This state of affairs is surprising given the importance of writing tasks in L2 development (Manchón and Vasylets 2019). To fill this research gap, this theoretical paper aims to conceptualize writing modality as a task complexity factor. In this paper, we specifically focus on two writing modalities: pen-and-paper writing and computer writing which involves touch typing on a keyboard. Another central construct in this study is task complexity, which, in line with Robinson (2001), we define as cognitive demands posed by a task on L2 learners' memory and attention resources. To pursue the aim of this article, we first start with comprehensive summary of previous adaptations of the psycholinguistic task-based models to writing. We then compare computer and paper writing in terms of their general characteristics, outline theoretical views which highlight the importance of modality in learning and performance, and examine the available empirical evidence. This will serve as a basis to substantiate our conceptualization of writing modality as a task complexity factor. We then proceed with

the methodological suggestions which will set the basis for the testability of our theoretical tenets. We finish with the theoretical, methodological and pedagogical implications of our proposal.

## 2. Application of Cognitive Task Complexity Models to L2 Writing

Much of the prominence in the task-based language teaching (TBLT) research has been gained by the psycholinguistic strand, whose main guiding frameworks have been the Cognition Hypothesis (Robinson 2001, 2011) and the Trade-Off Hypothesis (Skehan 1996, 2014). The two models make a number of predictions about how task characteristics can influence mental processing and performance of L2 learners. Skehan (1996), for example, predicts that, because of the limitations in cognitive resources, learners would trade off accuracy against complexity (or vice versa) in cognitively complex tasks. That is, in a complex task, a learner performance can be either accurate and fluent or complex and fluent, but not accurate and complex at the same time.

An alternative account of the effects of task demands has been offered by Robinson (2001, 2011) in his Cognition Hypothesis, which makes a number of predictions on the way task complexity can influence L2 output, interaction and involvement of individual differences in L2 performance and learning. To guide research into his claims, Robinson (2001) proposed an operational taxonomy which distinguishes between three dimensions of task characteristics: *task complexity*, *task condition* and *task difficulty*.

The *task complexity* dimension refers to the factors that contribute to the cognitive demands that tasks make on learners' memory and attention resources. Within this dimension, an important theoretical distinction is made between resource-dispersing and resource-directing variables, which can produce varied effects on learners' performance and learning. The prediction is that the increase of demands related to the resource-dispersing variables (e.g., by limiting planning time) would disperse/tax learners' attentional resources, adversely affecting accuracy, complexity and fluency of performance. At the same time, practice of tasks with increased resource-dispersing demands is considered to benefit faster and more automatic L2 access and use. On the other hand, increases of task complexity related to the resource-directing variables (e.g., by increasing reasoning demands) is expected to pose higher conceptual and linguistic demands. Such an increase in task complexity is theorized to enhance learners' noticing of L2 forms and mobilize their cognitive and linguistic resources, which is expected to benefit both L2 development and performance. In particular, it is expected that increased resource-directing demands would induce learners to produce more accurate and complex language, while negatively affecting fluency.

The second category in Robinson's taxonomy is the *task condition* dimension, which is divided into *participation* variables (e.g., the number of participants in the task, time on task) and *participant* variables (e.g., gender, proficiency level, etc.). The third dimension, *task difficulty*, relates to learners' individual differences, which include ability factors (e.g., working memory, language aptitude, etc.) and affective factors (e.g., motivation, anxiety, etc.). The prediction is that task difficulty factors may be responsible for between-learner variation when performing L2 tasks. Robinson also predicts that the role of task difficulty factors might be more prominent in complex tasks.

Although designed with the oral mode in mind, both Skehan's and Robinson's models have been applied to writing task performance as well. Neither Robinson nor Skehan ever explicitly stated their theoretical predictions should be applied exclusively to the oral mode of production. However, none of these theories discuss if and how their predictions could apply to L2 writing. This neglect of writing is surprising provided the importance of writing tasks in language acquisition (Manchón and Vasylets 2019), as well as difficulty of learning to write in an L2. Thus, considering that even in the native language, writing has been viewed as a complex skill to acquire (Bereiter and Scardamalia 1987), it is not surprising that L2 writing is characterized by a high variability in the attained levels of L2 writing proficiency (Weigle 2005). To become a proficient L2 writer, a learner needs to

acquire multiple skills and types of knowledge. Thus, in addition to L2 vocabulary and grammar, learners have to acquire the knowledge of L2 orthography. Additionally, genre and rhetorical requirements can differ in an L2 (Loi 2010), which can demand additional learning efforts on the part of L2 writers. Cognitive resources, such as working memory, are also recruited in L2 writing (Vasylets and Marín 2021). Thus, because of its importance and difficulty, L2 writing needs to occupy a more central role in TBLT theorizing and research. Recently, though, a number of studies have tested empirically the predictions of cognitive TBLT models as applied to L2 writing tasks. These studies have produced mixed results. Johnson (2017), for example, has reported that cumulative findings offered no clear support for the Cognition Hypothesis, although there seems to be an indication that task complexity may promote attention to the formulation and monitoring systems of the writing process. At the same time, Vasylets et al.'s (2017) comparative study of task complexity effects in L2 speech and writing reported a better fit to the Cognition Hypothesis predictions in writing. In particular, task complexity effects appeared to be more visible in the written mode, with the performance being more linguistically complex and accurate in the complex version of the writing task as compared to the simple counterpart task. At the same time, there was little difference between complex and simple oral task performance. This higher susceptibility of the writing tasks to task complexity manipulations was explained by the intrinsic conditions of writing, which allow for more self-paced and controlled implementation of language production processes as compared to hard-pressed speech. As a result, Vasylets et al. (2017, p. 421) suggested that written mode could represent "a perfect arena for the manifestation of task complexity effects".

There were also a number of theoretical attempts to adapt the existing task complexity models to writing. For example, Manchón (2014) assessed Robinson's Cognition Hypothesis for its suitability to account for L2 writing tasks. In this regard, Manchón suggested that the model would need revision in all the groups of task factors, including task complexity, task condition and task difficulty dimensions. Additionally, the major focus in this adaptation should be on planning, which represents "a uniquely distinctive phenomenon in writing" (p. 32), with more extensive planning possibilities being intrinsically available to writers. The dimension of task condition would also need rethinking, as the role of interaction in speech could be expected to work differently from writing, with speakers exchanging face-to-face messages and writers communicating with the audience, which is typically displaced in time and place. Finally, task difficulty dimension should incorporate some additional variables such as genre knowledge and L2 writing expertise.

Kormos (2014) has also offered her account of how Robinson's model can be adapted to writing, suggesting that modality (oral versus written) could be added as another task complexity factor. In this proposal, Kormos draws on the psycholinguistic comparison of speech and writing to justify that modality could fit as a task complexity dimension as it can pose both resource-directing and resource-dispersing demands. Because of the time-pressed conditions of online production, speech could be conceptualized as posing resource-dispersing demands; that is, posing the type of mental load which disperses learners' cognitive resources over different areas of performance. In other words, the real time nature of speech induces language users to devote their cognitive resources to various areas simultaneously, such as speech comprehension (understanding of the words of the interlocutor) and speech production processes (planning and linguistic formulation of the reply). On the other hand, writing is expected to pose resource-directing demands. That is, the self-paced nature of writing provides the conditions under which language users do not have to divide their attention between focus-on-meaning and focus-on-form. Learners, can, thus, direct their cognitive resources to a specific language process and/or performance area.

Writing itself, however, can take place in two basic writing modalities, namely pen-and-paper versus computer writing. Because of the intrinsic differences in their characteristics, paper and computer writing can pose differing demands on cognitive resources. We can consequently propose that writing modality can be conceptualized as a task complexity

factor. To substantiate our claims, we will first compare the general characteristics of the pen versus computer writing. Then, we will examine the role of modality in writing theories, which will be followed by the review of interdisciplinary empirical evidence showing that learning and performance can happen differently in paper versus computer writing. On this basis, we will defend the claim that writing modality can be seen as another task complexity factor.

### 3. Paper-Based versus Computer Writing: General Characteristics, Theoretical Framing and Empirical Evidence

#### 3.1. General Characteristics

The primary difference between paper and computer writing lies in the motor execution of the text (handwriting versus typing) and in the material media involved (pen-and-paper versus screen and keyboard/mouse).

Thus, paper writing is based on handwriting, which requires individuals to execute grapho-motorically each letter/symbol stroke-by-stroke. During this process, visual attention is also deeply involved with the focus being on the writing surface (i.e., paper). This makes handwriting a slow and laborious activity, which is highly embodied as it requires deep integration of cognitive and attentional resources with motor and perceptual skills (Mangen and Balsvik 2016). Moreover, by employing their own motor system to create every letter, writers experience a sense of ownership and are exposed to letter variability, as one handwritten letter may vary from another instance of the same letter (Feder and Majnemer 2007). Haptic interactions with paper and a writing device are also noticeable, which enriches kinesthetic experience. Fixed layout and the touchable/tangible nature of a paper page also support construction of a stable and efficient cognitive map (i.e., visual representation) of the written text (Hou et al. 2017).

Computer writing is based on typing, which requires discrimination among letters in the process of keys selection. Typing, thus, represents a form of spatial learning in which a writer has to create a cognitive map of the keyboard (Kiefer et al. 2015). The attention of writers has to be divided between keyboard and screen and, rather than crafting the letters using their own motor system, typists serve themselves to the ready-made symbols which simplifies motor functions (Mangen and Velay 2014). Visual experience with the written text is also different as the typed letters are uniform and the text appears on a screen which is intangible, detached and mediated (Mangen and Balsvik 2016). As a result, computer writing represents a writing modality which is more phenomenologically monotonous, impersonalized and disembodied, but which is also faster and less laborious as compared to paper-based writing (Mangen and Balsvik 2016; Mangen and Velay 2010).

In sum, paper and computer writing have important cognitive/attentional, sensory-motor, visual and haptic differences (Table 1), whose potential implications for language performance and learning cannot be ignored. The relevance of these differences is recognized in multiple theoretical perspectives, as will be shown in the next section.

**Table 1.** Differences between pen and computer writing.

| Pen-and-Paper Writing | Computer Writing |
| --- | --- |
| Pen | Keyboard/mouse |
| Paper | Screen |
| Stroke-by-stroke execution of visual signs | Selection of ready-made symbols |
| Attention on the writing surface | Attention on screen and keyboard |
| Variable letters/signs | Uniform letters/signs |
| Rich kinesthetic experience | Less varied kinesthetic experience |
| Personal and highly embodied experience | Detached experience |

#### 3.2. Theoretical Framing

From the theoretical standpoint, the role of modality in writing is recognized in multiple perspectives, including cognitive writing theory (Flower and Hayes 1980; Hayes

2012; Kellogg 1996; McCutchen 1996), semiotics (Kress 2003), embodied cognition (Clark and Chalmers 1998) and the related socio-cognitive theory of second language acquisition (Atkinson 2011).

Thus, the role of transcription processes is acknowledged in cognitive models of writing production. One example is the writing model by Flower and Hayes (1980). This seminal model originally comprised three main parts: (i) task environment, which includes writer-external factors, such as the type of written assignment, audience, and the text produced so far, (ii) writers' long-term memory, and (iii) the general writing process composed of *planning* (creation of conceptual content), *translating* of conceptual ideas into linguistic form and *reviewing*. This original model has been updated by Hayes (2012) with some substantial changes, such as the addition of the *transcription* process. As Hayes (2012) posits, "transcription does compete with other writing processes for cognitive sources in both adults and children and must be accounted for in modeling all writers" (pp. 371–72). Another important update was an incorporation of the element of *transcribing technology* within the task environment dimension. It must be admitted, though, that Hayes (2012) does not elaborate on the constituting elements of *transcribing technology* (which could include, for example, pen and computer processor). Another limitation is that Hayes does not make any theoretical predictions concerning the role of transcribing technology in performance or learning.

Kellogg's (1996) model of writing also includes *execution* (motor programming and physical realization of the message), which is defined as a low-order non-optional writing process. According to Kellogg (1996), execution poses lower cognitive demands as compared to formulation and revision. Cognitive writing research also acknowledges links between grapho-motor processes and higher writing cognition. Thus, according to the capacity theory (McCutchen 1996), automatization of transcription is expected to free up cognitive resources, making them available for high-level writing processes and strategies.

The importance of modality is also highlighted is semiotics. To illustrate this point, a major semiotics theoretician Günther Kress (2003) points out important changes that digital technology produces in writing:

> The combined effects on writing of the dominance of the mode of image and of the medium of screen will produce deep changes in the forms and functions of writing. This in turn will have profound effects on human, cognitive/affective, cultural and bodily engagement with the world, and on forms and shapes of knowledge. (Kress 2003, p. 3)

With this statement, Kress (2003) acknowledges the importance of the specific features of writing modality in learning, inter alia. Theoretical justification for the role of modality can also be found in the tenets of embodied cognition. Conceptions of embodiment take many forms (Barsalou 2008), but the main underlying idea in embodied cognition is that human cognition and learning, rather than being abstract and amodal, may instead be dependent on sensorimotor processing and interaction with the environment (Wilson 2002). In other words, the brain is not viewed as the only cognitive resource. Instead, cognition is seen as a combination of multiple resources, which include mind, body and their relations with the environment (Clark 2001; Wilson 2002). A major proponent of embodiment in second language acquisition (SLA) is Atkinson (2011), who introduced the socio-cognitive perspective as an alternative approach to explain L2 learning. The core claim of this approach is that "mind, body, and world function integratively in second language acquisition" (Atkinson 2011, p. 143). In this stance, SLA is viewed as a natural, dynamic process and cognition is reconceptualized as adaptive intelligence which projects into the world and uses multiple affordances to help learning. Although Atkinson (2011) admits that the socio-cognitive view is "new and undeveloped" (p. 162), he also stresses that this standpoint is open to the full range of possibilities and applications.

Writing also represent a highly embodied activity, as it is contingent on the interactions between internal cognitive processes, motor and perception processes and external environment (pen, keyboard, screen, paper) (Mangen and Balsvik 2016). For this reason, the

embodied view can usefully inform writing research. As pointed out by Mangen and Velay (2010), "the perspective of embodied cognition presents itself as an adequate and timely remedy to the disembodied study of cognition and, hence, writing" (p. 308). Research on embodied cognition is interdisciplinary and is supported by a variety of methodological strategies. Arguably, among the most relevant theories, which can be connected to the ideas of embodiment, and which could be (potentially) applied to writing, are the motor theory of speech perception (Liberman and Mattingly 1985) and the theory of multisensory learning (Shams and Seitz 2008).

Developed originally for the perception of spoken language, the motor theory of speech perception centrally claims that recognition of speech phonemes involves the motor system. As a support to this theoretical claim, there is now ample empirical evidence which has demonstrated that perceiving speech involves neural activity of the motor system. For example, Watkins et al. (2003) found that both while listening to speech and while seeing speech-related lip movements, the participants showed enhanced muscle activity in the tongue. The motor cortex is also involved in visual perception of written signs (Longcamp et al. 2006). We can find some precedents of employing the tenets of the motor theory of speech perception to frame writing experiments. As an example, this theory was used as one of the guiding frameworks in the study by Mangen et al. (2015) which explored whether writing modality (writing by hand, typing on laptop keyboard or touch typing on virtual keyboard) has an effect on word learning. The results of this experiment showed certain benefits of handwriting for word learning. This finding was explained by the fact that rich kinesthesia and complex motor processes inherent in handwriting contributed to a more solid memory trace[1].

Another embodied theory which can be applied to writing research is a theory of multisensory learning (Shams and Seitz 2008). Proponents of this view hold that the human brain operates and learns more optimally in a multisensory environment. Thus, unisensory learning is expected to be less powerful than multisensory learning. For example, it has been shown that humans recognize a voice better after audiovisual training (voice co-presented with video of the speaking face) as compared to training with voice alone (Von Kriegstein and Giraud 2006). By extension, we could also argue that differences in the sensory experiences (e.g., haptics) in paper- and computer-based writing can influence learning in these two modalities. Thus, materiality of pen and paper, which are touchable and tangible, afford rich sensorimotor experiences in paper writing. On the other hand, the haptic experience in computer writing is mainly restricted to the keyboard, while the interaction with the written text is detached and intangible as it is displayed on the screen. As a result, paper-based writing can be defined as a richer sensorial modality as compared to computer writing, with the potential implications for learning and performance. Multisensory theory of learning has been fruitfully applied to frame research comparing reading on paper versus screen (Hou et al. 2017). Provided that reading constitutes an important part of the writing process (e.g., reading is a basis of revision in writing), this sets a precedent in the use of multisensory theory in the research on writing modality.

In sum, the role of writing modality is acknowledged in multiple theories from a variety of scientific fields, including cognitive writing research, semiotics and embodied cognition theories (see Table 2). These theories have been usefully employed to inform interdisciplinary research on the role of writing modality in learning and performance.

**Table 2.** Tenets relevant for the theoretical justification of the importance of writing modality in performance and learning.

| Theory | Relevance for Writing Modality |
| --- | --- |
| Cognitive writing theories | Importance of the transcription process (Hayes 2012; Kellogg 1996; McCutchen 1996) Importance of transcribing technologies (Hayes 2012) |
| Semiotics | Use of screen changes functions of writing and affects the way we acquire knowledge (Kress 2003) |
| Embodied cognition | Mind, body and world function integratively in human cognition (Wilson 2002), including SLA (Atkinson 2011) Connection between motor system and language (Liberman and Mattingly 1985) Importance of rich sensory experience for high-quality learning (Shams and Seitz 2008) |

*3.3. Empirical Evidence*

3.3.1. Findings Form Experimental Psychology and Neuroscience

Multiple studies from experimental psychology have predominantly employed theories of embodied cognition as a guiding framework to investigate differences in learning outcomes in handwriting versus typing. Overall, these studies point to an advantage of handwriting over typing in letter learning (Longcamp et al. 2005, 2006), word learning (Mangen et al. 2015), as well as word writing and reading (Kiefer et al. 2015). In line with the embodied theories of action−perception, these results are typically explained by richer haptic−kinesthetic experience in handwriting, which enhances memory traces of letters and words with the consequent benefits for learning. The evidence for the importance of the motor component in writing also comes from studies which showed learning benefits of handwritten text copying. Thus, repeated writing by hand has proven to be a useful technique to help memorize kanji characters for Japanese schoolchildren (Naka and Naoi 1995) as well as for Japanese heritage learners (Kaho 2020). The importance of motor movements in language production can also be seen in the well-known phenomenon in Japan, commonly referred to as "kuuscho" (Sasaki 1987), which consists in writing with the finger in the air to identify and mentally retrieve a complex character.

Evidence on the variations in the mental processes and learning outcomes in different writing modalities also comes from neuroscientific approaches using electroencephalography (EEG; Ihara et al. 2021) and functional magnetic resonance imaging (fMRI; Vinci-Booher and James 2020). Embodied theories also constitute a common theoretical framework in this line of research (e.g., Askvik et al. 2020). Thus, the studies with fMRI techniques have demonstrated that variability of letter forms present in writing is beneficial for learning. Similarly, the study by Vinci-Booher and James (2020), which involved both children and adults, found evidence that handwritten letter variability may contribute to developmental changes in the neural systems that support letter perception. These results support findings for preliterate children by James and Engelhardt (2012), who concluded that handwriting may facilitate reading acquisition in young children.

A recent EEG study by Askvik et al. (2020) reported that, as compared to typing, handwriting with digital pen was associated with increased activation in the brain areas important for memory and for encoding of new information. Using the EEG technique, Ihara et al. (2021) has found increased word learning by means of handwriting (with either ink or digital pen) as compared to word learning using typing. Additionally, the participants reported more positive mood in the handwriting condition as compared to typing. Another EEG study by Osugi et al. (2019) compared outcomes of learning to read difficult words by writing with an ink pen versus a digital pen. In this study, familiarity with use of the digital pen played a role, as word learning was greater with a digital pen in the group which was familiar with its use (see also, Mangen et al. 2015; Ouellette and Tims 2014).

In sum, findings from experimental psychology and neuroscience have predominantly shown higher learning gains in handwriting as compared to typing, which has been typically attributed to richer haptic−kinesthetic experience in handwritten production. These findings are in line with the studies that have shown positive associations between

good handwriting skills and different dimensions of academic skills (Dinehart 2015). These studies, however, represent highly controlled experiments focusing on the learning of discreet linguistic features, such as letter or words. For this reason, it is vital to examine the results of the modality investigations which involved production of longer texts.

3.3.2. Findings from L1 and L2 Writing Research

In L1 and L2 writing studies, cognitive writing theories have been employed as an explanatory framework. The results of these investigations have been mixed. Thus, overall, cumulative findings seem to indicate that the texts written on the computer tend to be longer and display higher quality (in terms of language use, argument development, etc.) as compared to the pen-and-paper texts (Cheung 2012, 2016; Lam and Pennington 1995; Lee 2002, 2004; Li and Cumming 2001). Other studies, however, have reported absence of differences in the quality of writing in the two modalities (Chan et al. 2017; Kohler 2015; Weir et al. 2007).

Mixed results have also been found for the writing processes. Thus, while some studies have found that computer writing led to more effective higher-level revisions (Li and Cumming 2001), other studies reported that when writing on the computer, students revised more at a superficial level (Joram et al. 1992). Interesting insights were obtained by Chan et al. (2017), whose interview data revealed that the participants were more relaxed during their initial planning and they felt more comfortable making changes in computer writing. On the other hand, participants were reluctant to make changes in the handwritten texts, which resulted in more careful linguistic formulation (lexical searchers, in particular) in paper writing. Similarly, the study by Zhi and Huang (2021) showed that writers planned and revised more when writing on the computer.

Caution, however, must be taken when making definitive conclusions about writing process and product differences in the two modalities. In this regard, a number of criticisms were raised concerning some methodological problems of some previous studies, such as lack of a clear definition of how L2 proficiency was measured (e.g., Li 2006; Li and Cumming 2001) or lack of the information about time on task or use of external sources (e.g., Li 2006). Absence of this important information makes it problematic to compare findings or explain discrepancies in the research findings.

Inconsistent findings in previous research may also be explained by the lack of the control of some important mediating variables, such as writers' handwriting skills and computer literacy. Barkaoui (2016), for example, showed that writers with higher typing skills engaged in more planning, organizing and revising activities as compared to less skillful typists. Typing skills can also predict quality of writing texts. Thus, the study by Zhi and Huang (2021) showed that typing speed positively correlated with the holistic and analytic scores (e.g., task achievement, grammar) of computer written texts. Similarly, handwriting skills can also mediate quality of manually written texts (Graham et al. 2000).

Another important variable which has to be controlled for is computer familiarity (Ihara et al. 2021). Thus, Chan et al. (2017) reported that writers' familiarity with the computers positively correlated with the writing quality of computer written texts. Chan et al. (2017) also found that their participants were more familiar and comfortable with using computers than participants in earlier studies (e.g., Weir et al. 2007). This increase in computer familiarity in the recent years can be one of the explanatory factors to account for the discrepancies in the results of early and more recent investigations. Another relevant variable to control for is authenticity (i.e., correspondence to the real-life writing behaviors) of the writing modality. Thus, the participants in the study by Zhi and Huang (2021) found the computer-based writing modality to be more authentic. On the other hand, inconveniency of the revision process in paper writing induced the participants to modify their writing behaviors.

In sum, previous studies from different research areas provide evidence that paper-based and computer writing may engage learning and writing processes differently, which has potential consequences for writing performance and learning. Findings from ex-

perimental psychology and neuroscience seem to indicate that richer haptic−kinesthetic experience in paper writing may favor letter/word learning processes. At the same time, recent writing research has provided evidence that computer writing might favor writing processes and performance (see Table 3). On the basis of these results, we can conclude that the affordances of paper-based and computer writing can have an impact on learning/writing processes and writing performance. This provides empirical evidence for our thesis that writing modality can be defined as a task complexity factor.

**Table 3.** Summary of some relevant empirical findings for pen-and-paper and computer writing.

| Pen-and-Paper Writing | Computer Writing (Typing) |
| --- | --- |
| Benefits for learning to spell, letter/word learning | Higher text quality |
| More careful linguistics formulation | More planning and revision |
| Increased activation of the brain areas responsible for memory and learning | Higher perceived authenticity |
| Hand-copying of texts benefits language learning | Higher computer literacy benefits writing processes/products |
| Handwriting skills mediate writing processes/products | Typing skills mediate writing processes/products |

### 4. Conceptualization of Writing Modality as a Task Complexity Factor

The important differences in haptic−kinesthetic affordances in the two writing modalities may cause writers to allocate their cognitive resources differently in order to successfully meet the linguistic demands of tasks in paper versus computer writing. In other words, it could be suggested that writing modality might be conceptualized as a task complexity factor, which, according to Robinson (2001), constitutes a factor contributing to the cognitive load tasks make on learners' attention and reasoning. As mentioned previously, within the task complexity factor an important distinction is made between cognitive/conceptual (resource-*directing*) and performative/procedural (resource-*dispersing*) dimensions. Both dimensions are relevant for learning, as the increased task complexity along the resource-directing dimension is believed to benefit analysis and development of interlanguage, while enhanced resource-dispersing demands are posited to promote access to and automatization of interlanguage (Robinson 2001, 2011). This distinction is similar to the one made by Bialystok (1994), who differentiated between the *analysis* and *control* in L2 learning.

Based on our previous comparison of paper and computer writing, it would be feasible to hypothesize that writing modality can pose both attentional resource-dispersing and resource-directing demands. We would also like to suggest that the type of cognitive load (dispersing or directing) exerted by a writing modality would essentially depend on learner characteristics. Thus, for writers who are not experienced typists (for example, young learners), performance of a writing task on the computer could pose resource-dispersing demands, as the writers' cognitive resources would have to be shared with the high- and low-order (i.e., execution) processes. On the other hand, experienced typists, with automatized execution process in the digital environment, would be able to direct their cognitive resources primarily to high-order processes of planning, formulation and revision. Concerning paper writing, its slowness and rich haptic−kinesthetic experience can account for deeper processing during task performance, with the concomitant direction of the cognitive resources to the learning and writing processes. At the same time, lack of familiarity with paper writing can disperse learners' resources.

Another way to conceptualize writing modality as a task complexity factor would be to define paper and computer writing as tasks which would differ in the amount of cognitive load they pose on a particular learner. Our suggestion is that paper writing could represent a simple task, while computer writing could be defined as complex. Importantly, the reverse could also be true. Our thesis is that the definition of the simple and complex tasks in the context of writing modality could be tied to the learner characteristics, such age or L2 proficiency, but the factor of the central importance would be writing experience. Thus, for

learners who have more experience with computer writing, this particular modality would represent a simple task, while writing on paper would pose greater cognitive load, and, thus constitute a complex task. Conversely, the experience of other learners may convert paper writing into the simple task and computer writing into a complex one.

Writing experience (previous writing training, years of experience, knowledge of writing genres, daily time spent on writing activity, etc.) is crucial as it can determine the levels of fatigue, immersion and perceived authenticity experienced by writers when writing in different modalities. Measures of fatigue are common in research comparing reading on screen versus paper (Hou et al. 2017). By the same token, writing on paper versus using keyboard/screen could be more or less fatiguing, and, thus, aggravate or relieve the cognitive load put on the mental resources. Immersion, defined as a sense of engagement or an experience of losing oneself in an environment (Witmer and Singer 1998), is also a relevant variable. Studies on reading, for example, have shown the levels of immersion could be different in book and screen reading, which have consequences for how much information the participants learned from the text (Mangen et al. 2013; Støle et al. 2020). Similarly, writing on paper versus on computer could produce different levels of immersion and induce writers to enter a lower or higher level of writing flow, with concomitant consequences for the level of the posed cognitive load (Kellogg 1999). Familiarity with the modality can also be relevant (Chen et al. 2014), as well as the perceived authenticity (i.e., naturalness or real-life resemblance) of the writing modality. For example, Zhi and Huang (2021) have found that writing in a less authentic mode can impose additional cognitive load and induce changes in the writing processes and outcomes.

In sum, individual differences (centrally, writing experience) could determine the type and/or amount of cognitive load posed by the writing modality during task performance. For this reason, in order to determine if paper writing represents a cognitively simple task as opposed to computer writing or vice versa, it is crucial to measure cognitive load empirically.

## 5. Methodological Implications: Validation of Cognitive Load in Paper-Based and Computer Writing

One way of tapping into the cognitive load posed by writing modalities could be to examine the nature of L2 writing processes and/or the quality of output in simple versus complex writing tasks. However, inferring cognitive task complexity from the observations/comparison of L2 processes/performance could lead to circularity. Thus, in order to get beyond this circular reasoning, we need to employ independent measures to measure task complexity (Révész 2014).

In the area of TBLT, researchers have employed various measures of task complexity, including learner self-ratings (Baralt 2013), expert judgments (Révész 2014; Révész et al. 2014, 2016, 2017), stimulated recall and interviews (Kim 2009), time-on-task (Vasylets 2017; Lee 2020), time estimation (Sasayama 2016), dual-task methodology (Xu et al. 2021) or eye tracking (Révész et al. 2014). In our view, many of these techniques could also be applied to assess the cognitive load posed by paper versus computer writing.

So far, the most common method to verify task complexity has been the use of subjective self-ratings, which typically ask learners to judge the experienced level of cognitive load on a Likert-scale ranging from "no mental effort at all" to "extreme mental effort". The underlying premise of this method is that people have a capacity to assign a numerical value to the amount of mental effort they spend during a particular cognitive activity (Brünken et al. 2010). Studies in psychology have shown that self-rating is a valid, reliable and unintrusive method, which is sensitive to relatively small differences in cognitive load (e.g., Ayres 2006). Easiness in administration and analysis have made self-rating popular in TBLT research (Révész et al. 2016; Sasayama 2016). The most recent example of the application of self-rating to writing can be found in Xu et al. (2021), who established a 9-point Likert scale with two items judging: (1) the mental effort required by the task (1 = "this task required no mental effort at all"; 9 = "this task required extreme mental

effort"), and (2) overall task difficulty (1 = "this task was not difficult at all"; 9 = "this task was extremely difficult"). In this scale, the question about mental effort was intended as a direct measurement of cognitive demand, reflecting "the amount of resources actually allocated to accommodate the task demands" (Paas et al. 1994, p. 420). Thus, mental effort is considered to reflect the actual effort engaged by the individual in task performance. On the other hand, the task difficulty question is considered to tap into the supposed demand imposed by a task, thus, focusing on the individual rather than on the task itself. A task which is perceived as difficult does not guarantee that higher mental effort will be involved and vice versa.

Other instruments which could be employed to measure cognitive load in different writing modalities are stimulated recall and interviews. These techniques have been extensively used in in the exploration of L2 writers' cognition (Hort and Vasylets in press). Stimulated recall represents a recall session which intends to tap into the thoughts the learners had during writing (Gass and Mackey 2016). As stimulated recall takes place after the completion of the writing task, the technique is not reactive, i.e., it does not alter the natural flow of the writing processes. A potential disadvantage, though, is the issue of veridicality and information loss as writers may forget some details of the writing task (Bowles 2018; Olive 2010). To combat this issue, the produced written texts can be employed to stimulate memory during the recall session. Recent examples of the use of this technique in the TBLT writing research can be found in Révész et al. (2017), who combined stimulated recall with self-ratings to verify task complexity.

An encouraging approach to assess the cognitive load is the method of subjective time estimation (Fink and Neubauer 2001). This technique consists in asking the participants to estimate the duration of time-on-task in the absence of an external timing device. This technique is premised on the attentional model of time perception by Thomas and Weaver (1975), which posits that attention is shared between temporal and nontemporal information processing. With increase in the cognitive load of a nontemporal task, less attention is left to process temporal information. Consequently, the subjective estimation of task duration becomes less accurate. Time estimation has been successfully employed by Baralt (2013) and Sasayama (2016) with oral task performance. However, to our knowledge, there are no precedents of use of time estimation to assess the cognitive load of writing tasks. Thus, it is a matter for future research to validate this promising technique in the domain of written task performance.

In addition to the subjective self-report measures, dual-task methodology could also be adapted to assess cognitive load in writing tasks of different modalities (Olive 2010). Dual-task methodology involves carrying out a primary task (e.g., composing a written text) simultaneously with doing a secondary task (typically a simple activity, such as detecting audio or visual stimuli). The assumption is that individuals have to share their limited cognitive resources between primary and secondary tasks. Therefore, as the cognitive demands of the primary task increases performance in the secondary task is expected to decrease. Slower and/or less accurate secondary task performance is considered to be indicative of a higher cognitive load imposed by the primary task. In writing research, the dual task method has been extensively employed for various purposes, such as exploration of writing processes or testing the role of working memory in writing (Olive 2010). In the context of TBLT, studies by Révész et al. (2016) and Sasayama (2016) were the first to employ this method to assess cognitive demands of oral tasks. A recent study by Xu et al. (2021), which employed auditory stimulus detection as a secondary task during the primary task of writing, has created a precedent for the use of the dual task method to compare cognitive load posed by pen versus computer written tasks. Although the administration of this technique is more time-consuming as compared to self-ratings, an advantage of the dual task method is that it measures cognitive load concurrently and, thus, avoids problems of memory decay or veridicality inherent in the post-task retrospective methods.

Finally, we could also consider the use of physiological techniques to assess task complexity in writing. The underlying assumption is that changes in cognitive functioning

have physiological effects, such as changes in heart rate activity, brain activity (e.g., task-evoked brain potentials) and eye activity (e.g., pupillary dilation, duration of eye fixations). While heart rate variability has proven to be insensitive to subtle fluctuations in cognitive load (Paas et al. 1994), the eye-tracking technique seems to hold promise for the exploration of the cognitive load posed by paper and computer-based writing. The eye-tracking technique basically detects and measures an eye's movements (saccades), stops (fixations), as well as movements back (regressions). Because of the connection between eye gaze and attention, known as the "eye-mind principle" (Reichle et al. 2006), eye movements have been employed as a quantitative measure of attention and processing in SLA (Conklin and Pellicer-Sánchez 2016). The few available L2 writing studies have triangulated eye-tracking measures with other techniques to explore L2 writing fluency (Chukharev-Hudilainen et al. 2019) or L2 writing processes (Gánem-Gutiérrez and Gilmore 2018). To our knowledge, to date only Révész et al. (2014) have employed eye tracking to assess the cognitive load of oral tasks. In this study, the designed-to-be complex version of the tasks elicited more and longer fixations, which confirmed the initial hypothesis. In our view, this previous research provides a basis for the use of eye tracking as a validation technique of cognitive load posed by pen and computer writing. The most obvious way to do it would be to focus on the reading/revising stage of the writing, the interpretation of which could be informed by the findings from writing studies and also from numerous eye-tracking studies on L1/L2 reading (Dirix et al. 2020). Finally, self-rating scales could be employed to assess fatigue, immersion, familiarity and perceived authenticity of the writing modalities (Chen et al. 2014; Hou et al. 2017; Zhi and Huang 2021).

In sum, various measures can be employed to independently assess the cognitive load imposed by writing tasks (see Table 4). This assessment is crucial as it will help verify empirically if and how the two modalities differ in terms of the level of cognitive complexity for different types of learners.

**Table 4.** Measures of cognitive load in writing tasks.

| Self-Report Measures | Behavioral Measures | Physiological Measures | Other Measures |
| --- | --- | --- | --- |
| Self-ratings (Likert-scale) | Time-on-task | Brain activity (EEG) | Expert judgments |
| Stimulated recall | Dual task methodology | Eye activity (eye tracking) | |
| Interviews | | | |
| Time estimation | | | |

## 6. Conclusions, Implications and Future Research

In this paper we have advanced a proposal that writing modality (paper-based versus computer writing) can be conceptualized as a task complexity factor. Our starting point was to examine theoretical views which underscore the importance of writing modality. In addition, we looked into the interdisciplinary empirical studies which examined potential differences between paper-based and computer writing. This provided us with a theoretical and empirical justification of our hypothesis that these two types of writing can differ in terms of their L2 learning and performance affordances. Figure 1 summarizes our main ideas. Thus, we propose that writing modality can pose both resource-directing and resource-dispersing cognitive demands. The amount/type of cognitive load posed by the writing modality is determined and/or mediated by learner individual differences, with writing experience presumably being one of the key characteristics. Our thesis is that, depending on their age, skills and knowledge, learners may experience different levels of fatigue, immersion or perceived authenticity in a particular writing modality, which will also impact the type/amount of the cognitive load posed by the writing task. For this reason, we consider it of central importance to empirically assess cognitive load, as well as other relevant variables (e.g., writing expertise, L2 proficiency) for each type of learner within a particular writing context. As such, writing modality appears as a highly learner-sensitive task complexity factor. According to Robinson's theory, one of the main

effects of task complexity reveals itself in complexity, accuracy and fluency of performance. For this reason, the comparison of writing quality in pen versus computer writing would also contribute to our understanding of writing modality as a task complexity dimension.

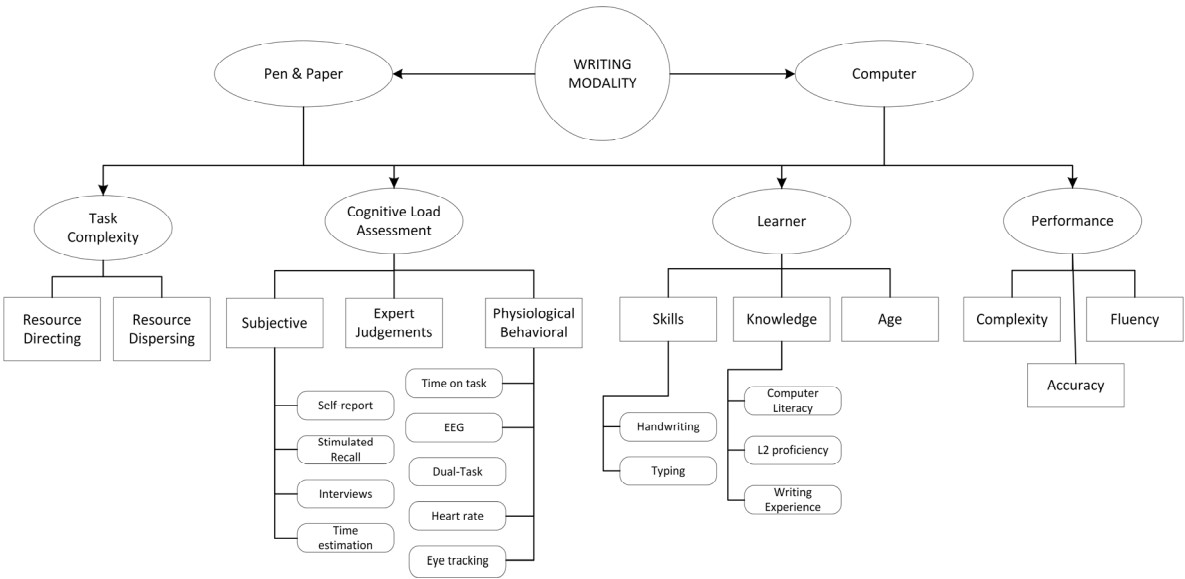

**Figure 1.** Writing modality as a task complexity factor.

It is important to highlight, however, that the tenets that we advance will remain tentative until they undergo empirical verification in both experimental and classroom-based studies. An example of this type of investigation could be a study with a within-learner design in which the participants would carry out, in the counterbalanced order, several L2 writing tasks in both pen and computer writing modalities. After the performance of each writing task, the participants would complete a Likert-scale questionnaire tapping into the self-perceived cognitive load, task motivation/engagement, task authenticity, immersion and fatigue. The participants' writing expertise and L2 proficiency should also be controlled. In the final phase of the experiment, the participants would be interviewed to gain further insights on their perception of the level of task complexity in pen versus computer writing. To measure the quality of L2 writing performance, the obtained written output would be assessed for complexity, accuracy, fluency and communicative adequacy of performance (Vasylets et al. 2020). In the analysis, quality of handwritten versus computer written texts would be compared, and the data from the questionnaires and interviews would allow comparison of the levels of self-perceived cognitive load, task motivation, immersion and fatigue when writing on paper versus computer. The results of this study would help elucidate which task (handwritten or computer written) is perceived by learners as being simple or complex. Additionally, the study would contribute to the knowledge about the quality of L2 writing performance using a pen vs. computer writing.

Finally, we consider that our theoretical conceptualization of writing modality as task complexity factor may bear important theoretical and pedagogical implications. On the theoretical plane, incorporation of our proposal into the general tenets of the Cognition Hypothesis would mean a greater approximation of task complexity theorizing to the embodied views on human learning and performance. In our opinion, an important advantage of the embodied cognition is that, rather than treating learners as merely a computational system, this view adds to the equation the external environment and body affordances. In other words, language learning is viewed as contingent on the way we interact with the outer world and engage our body in language production. It is our belief that, by embracing embodiment, we would gain a deeper understanding of writing task performance as a tool to learn a language. In terms of pedagogical implications, knowledge about the idiosyncrasy of learning/performance in different writing modalities is important

for syllabus designers and teachers. This knowledge will enable the stakeholders to achieve the optimal task sequencing and implementation, which would ultimately lead to the enhancement of the learning results.

Concerning future research, we would like to underscore once again the importance of the empirical verification of the cognitive load of tasks in different writing modalities, as well as measurement of other relevant variables such as task immersion or authenticity of the writing process. Numerous methods are available for this end, and it is a matter of future research to determine the most efficient and practical method to employ in the context of writing performance. Future research should also focus on the interactions between writing modality and task condition factors, such as time-on-task or the number of the participants (individual versus collaborative performance). Interactions between learner individual differences and writing modality should be further explored. Pioneering studies in this issue have reported that the effects of affective (Vasylets and Mellado 2022) as well as cognitive traits (Vasylets et al. in press) may differ in paper and computer writing. However, more research efforts in this regard are required.

**Author Contributions:** Conceptualization, O.V. and J.M.; writing—original draft preparation, O.V. and J.M.; writing—review and editing, O.V. and J.M. All authors have read and agreed to the published version of the manuscript.

**Funding:** The research reported in this paper is part of the work conducted within two research projects on L2 writing financed by the Spanish Ministerio de Economía y Competitividad (research grant FFI 2016-79763-P) and by Fundación Séneca (research grant 20832/PI/18).

**Institutional Review Board Statement:** Not applicable.

**Informed Consent Statement:** Not applicable.

**Data Availability Statement:** Not applicable.

**Conflicts of Interest:** The authors declare no conflict of interest.

## Note

[1] As it is well-known, motor speech perception theory was developed to account specifically for human phonetic perception. The main postulates of this theory deal with the role that inner motor representations play in the processing of language. Thus, the viability of applying motor speech perception theory to account for written production needs further theoretical development and empirical testing.

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
