# Peer review of "Pen-and-Paper versus Computer-Mediated Writing Modality as a New Dimension of Task Complexity"

_languages, doi:10.3390/languages7030195_

Round 1

Reviewer 1 Report

Suggestions:

It is missing a () at the beginning of the abstract;

I suggest to erradicate the all reference to Nietzche considering the theme and the scope of this Journal  (it does not fit well until line 37);     

I highly recommend a section for L2 writing only because the L2 is well mentioned at the beginning of the paper and i felt that it is missing as concept and practice. Also, regarding the L2 writing, it may be positive to identify L2 in the title.

Reviewer 2 Report

Dear Colleagues:

Thank you for your submission, “Pen-and-paper versus computer-mediated writing modality as a new dimension of task complexity 2.” Your work aligns with my own teaching and research interests, and I appreciate how you offer new insights on the ways that we might consider writing modalities (handwritten/computer-mediated) and the effects that these will have on learners at various stages in the writing task. I can certainly appreciate your quest to better understand what is going on in our brains as we write, and I think of that both as a writer and as a teacher of writing. So, I agree that your ideas have merit and that a theoretical examination of this topic is warranted.

However, there are some concerns that I have about the overall clarity of your argument, and the level of detail that you provide to your reader. I have noted these concerns in the extensive comments listed below, and in the attached PDF. In short, I am recommending your manuscript for a “reconsider after major revision,” and I will document a few suggestions for revision that I hope you consider. I’ve broken them into some minor concerns and others that are more significant.

In terms of minor concerns:

  • There are many places with typographic errors or omissions. Please proofread carefully, as I am doubtful that I caught everything, though I did try to be thorough in what I noted for corrections.
  • Be sure to offer examples of what you mean in various places rather than relying on the reader to infer what you mean. I have indicated many spots that could use elaboration.
  • Some of the citations for writing studies, while foundational, also seem out-of-date. I am not an expert in psychology, so I am not sure about the relevance/currency of those references. Please do work to get the most up-to-date ones (for instance, a 1990 reference on spelling ability in relation to handwriting and computer-mediated writing does seem foundational, though certainly not current given that you are arguing that people’s writing habits are changing).
  • On a more substantive level, please provide some periodic summaries or “sign posts” for your readers, capturing your ideas and building from one section/idea to the next. For instance, summarize ideas related to task complexity, and then layer on ideas about cognitive load, and then perhaps more about embodiment, and so on. That is, build your model as outlined below.
  • Finally, do you want to bring the Nietzche story full circle? It serves as an interesting introduction, but then really does not come back into play.

Given those minor concerns, my major concern is that, in the end, you have laid out a list of items that we might consider, yet don’t provide a clear model, nor a call to action for researchers. I’ve tried to document places in your manuscript where this could be elaborated upon and, by the end, you should paint a clear picture for your readers in terms of what you want them to know and be able to do with the connections that you are drawing from various perspectives (task analysis, cognitive load, etc).

As one recent example of what you could conceivably emulate in terms of a mentor text, I strongly encourage you to review how Coiro has done this in her “Multifaceted Heuristic of Digital Reading,” in which she also builds a concept map showing the connections amongst all her ideas. So, please look at this as one example of how you might provide your readers with a clear, yet nuanced and rich model for what you are proposing in relation to writing modalities, task complexity, and cognitive load. Also, if this really is meant to be about L2 learners, it would seem that you need to layer these ideas in as well, especially give the focus of the special issue.

Coiro, J. (2021). Toward a multifaceted heuristic of digital reading to inform assessment, research, practice, and policy. Reading Research Quarterly, 56(1), 9–31. https://doi.org/10.1002/rrq.302

Overall, your manuscript is promising. You point to a key idea about the nature of writing with pen/cil and paper as compared to using digital means, and that deserves attention from all of us, as teachers and as researchers. My hope is that the feedback here will help move you toward an even more complete and engaging manuscript. 

Thank you for sharing your work and best of luck with your revisions.

Annotation summary:

--- Page 1 ---

Caret, Reviewer:
)

Caret, Reviewer:
measure of

Caret, Reviewer:
an individual’s

Strikeout (red), Reviewer:
started to fear

Caret, Reviewer:
feared

Strikeout (red), Reviewer:
Rather s

Caret, Reviewer:
S

Strikeout (red), Reviewer:
one

Caret, Reviewer:
some

Strikeout (red), Reviewer:
pen-and-paper

Caret, Reviewer:
paper-based

Strikeout (red), Reviewer:
based

Caret, Reviewer:
mediated

Strikeout (red), Reviewer:
if

Caret, Reviewer:
is

Caret, Reviewer:
,

Strikeout (red), Reviewer:
L2

Highlight (yellow), Reviewer:
Please offer definition here for “task complexity factor.”

Highlight (yellow), Reviewer:
Please complete this citation

Caret, Reviewer:
in XXXX,

Can you provide an approximate date for this?

Strikeout (red), Reviewer:
the

Caret, Reviewer:
a

Highlight (yellow), Reviewer:
Please stay consistent with your terms. Not that I disagree that “digital” could be compared with “computer mediated,“ or that “paper“ could be a substitute for something like “pen and paper,” but you need to remain consistent.

Strikeout (red), Reviewer:
(i.e., digital)

Highlight (yellow), Reviewer:
Having skimmed your entire manuscript, I know you get to this a bit later. However, here, can you please define what you mean by “modality”?

--- Page 2 ---

Highlight (yellow), Reviewer:
Please defined this acronym here at this first use.

Highlight (yellow), Reviewer:
You might consider enclosing this in parentheses

Caret, Reviewer:
‘s

Caret, Reviewer:
Perhaps add “likely” here?

Strikeout (red), Reviewer:
of

Caret, Reviewer:
about

Strikeout (red), Reviewer:
task

Caret, Reviewer:
a task’s

Highlight (yellow), Reviewer:
I understand what you mean here, yet the word “noticing” does not seem like quite the right word choice. Perhaps “attention to” or something similar?

Strikeout (red), Reviewer:

Highlight (yellow), Reviewer:
Perhaps the word “should” might be more fitting here?

Strikeout (red), Reviewer:
the

Caret, Reviewer:
Robinson’s

--- Page 3 ---

Caret, Reviewer:
the

Highlight (yellow), Reviewer:
This phrase has not been introduced yet, and could certainly use a little more elaboration.

Highlight (yellow), Reviewer:
Before moving on to say that “these ideas” resonate with other elements of the research literature, can you briefly summarize two or three main points? Perhaps a brief bulleted list would be helpful to create a concise summary here?

Strikeout (red), Reviewer:
on

Highlight (yellow), Reviewer:
This is unclear. What do you mean by “dimension of task dimension”? Please elaborate.

Highlight (yellow), Reviewer:
This is at least the second time that you’ve mentioned the differences between oral and written language processing. Made it behoove your argument, earlier, to lay out some of the major tenants that they share as well as differences between the two as well?

Highlight (yellow), Reviewer:
Again, you are going to want to come to terms much earlier on with definitions for “mode” and “modality.”

Highlight (yellow), Reviewer:
The thought in the segment of the sentence seems to trail off and is incomplete.

Highlight (yellow), Reviewer:
Do you mean here to say “on the other hand”?

Highlight (yellow), Reviewer:
Here is another term that needs to be defined more clearly, and much earlier on.

Highlight (yellow), Reviewer:
I will only make note of it one more time here, or else I will just be repetitive. Please stay consistent with the ways in which you describe computer – mediated and paper-based writing.

Highlight (yellow), Reviewer:
Can you provide a brief list of some of these avoidances as a parenthetical remark?

Highlight (yellow), Reviewer:
Again, providing a brief list of these cognitive demands will help situate your argument.

Caret, Reviewer:
various

Strikeout (red), Reviewer:
form

Caret, Reviewer:
form

Highlight (yellow), Reviewer:
This might need to be hyphenated.

Strikeout (red), Reviewer:
al

Strikeout (red), Reviewer:
attentional

Highlight (yellow), Reviewer:
I’m not sure that I 100% agree with this, unless you are talking about very young children. There is a level of automaticity that writers gain certainly by the time they are an upper elementary school such that they do not need to continually watch the end of their pen while they are writing words on paper.

Note (yellow), Reviewer:
Before moving forward, I wonder if the reader would benefit from having a brief summary of the major points from the section, perhaps included in a table as a way to clearly define all elements of the task dimension?

--- Page 4 ---

Caret, Reviewer:
, at least in relation to touch typing

Highlight (yellow), Reviewer:
I’m not sure about the word choice here. Perhaps “present“ or “noticeable” instead?

Highlight (yellow), Reviewer:
Just offering a stylistic note that you use this phrase – “on the other hand” – quite often. It is usually presented on a tone, without in “on the one hand” to explicitly set up a contrast.

Highlight (yellow), Reviewer:
Building on the suggestion from the end of the last section, I wonder if a brief table here could be helpful, summarizing these main points? Or, perhaps later in the manuscript, having all of the items summarized together could be helpful, too?

Strikeout (red), Reviewer:
Haye`s

Caret, Reviewer:
Hayes’

Caret, Reviewer:
Please add a space here

Caret, Reviewer:
does

Strikeout (red), Reviewer:
s

Caret, Reviewer:
,

Note (yellow), Reviewer:
Can you posit, then, what he might mean by transcription technology?

--- Page 5 ---

Highlight (yellow), Reviewer:
For some reason, the word “thus” does not feel like the right transitional phrase here. I’m not sure what you might use otherwise, perhaps “to the end” or “to illustrate this point”?

Strikeout (red), Reviewer:
Please either elaborate on the other things that you are mentioning here, or delete this phrase.

Caret, Reviewer:
the

Caret, Reviewer:
,

Highlight (yellow), Reviewer:
Please fill out this acronym on first use.

Caret, Reviewer:
Please include a space after the period. In fact, this has happened at many instances throughout the manuscript, and I will stop marking it here. Please just be mindful and check for this in later citations.

Highlight (yellow), Reviewer:
Please elaborate and explain why.

Caret, Reviewer:
s

Highlight (yellow), Reviewer:
Here, do you mean “views” in a plural sense, or “view” in a singular sense?

Caret, Reviewer:
,

Strikeout (red), Reviewer:
form

Caret, Reviewer:
from

Caret, Reviewer:
.

Highlight (yellow), Reviewer:
It is not entirely clear how this connects to your previous point about the tools. Please clarify.

--- Page 6 ---

Caret, Reviewer:
the

Highlight (yellow), Reviewer:
Can you please elaborate on what the specific differences might be?

Highlight (yellow), Reviewer:
Again, here is another place where a table or bulleted list summarizing your main ideas could be helpful.

Caret, Reviewer:
,

Highlight (yellow), Reviewer:
This doesn’t make sense here, and perhaps you mean “copying?”

Highlight (yellow), Reviewer:
This feels like it could use a little more elaboration.

Highlight (yellow), Reviewer:
So, this is interesting. I think it is the first time in the manuscript where you have documented other ways in which text can be produced in the computer, with the other example notably being voice – to – text dictation. It is likely that you might want to make it clear that, in nearly every instance, you are specifically talking about touch typing on a keyboard.

--- Page 7 ---

Strikeout (red), Reviewer:
y

Caret, Reviewer:
and

Highlight (yellow), Reviewer:
It feels as though you are transitioning to a significant new idea, and perhaps a subheading would be warranted here.

Highlight (yellow), Reviewer:
Higher quality in terms of what, specifically?

Strikeout (red), Reviewer:
digital

Caret, Reviewer:
computer-mediated

Caret, Reviewer:
,

Caret, Reviewer:
,

Highlight (yellow), Reviewer:
Again, the word “thus” doesn’t feel like quite the right transition here. You are trying to draw attention to the fact that we should take caution when considering these implications, so perhaps you want to use “however“ or something similar instead?

Highlight (yellow), Reviewer:
It’s probably not too big of a deal, but the double set of quotation marks feels a bit odd here, with examples embedded within examples.

Highlight (yellow), Reviewer:
I’m not entirely sure what you mean by this, though it seems to imply that people, especially children, are becoming more and more familiar with computer technologies in recent years and, by extension, are becoming more fluent with using them. Is this what you were trying to say/imply?

--- Page 8 ---

Highlight (yellow), Reviewer:
Just noting that you will want to stay consistent terms.

Highlight (yellow), Reviewer:
I’ve been struggling with the use of this word “empirical“ throughout your entire main script. In order to say that something is “empirical,“ it would suggest to me that you would be doing some kind of systematic study, most likely a controlled experimental design. I’m not quite sure what to say about whether or not you should continue using the word empirical, as it seems like you are making more of a theoretical argument instead.

Highlight (yellow), Reviewer:
It would be helpful to reiterate this definition here before moving forward.

Highlight (yellow), Reviewer:
Here, again, it would be useful to provide a few specific examples. Perhaps you could do this in parenthetical remarks, or you could include a table outlining the similarities and differences across reading modalities and the resource allocations.

Strikeout (red), Reviewer:
,

Caret, Reviewer:
er-

Highlight (yellow), Reviewer:
I don’t think that you have yet defined “cognitive load“ and you would want to do that in relation to “complexity factors“ as well.

--- Page 9 ---

Highlight (yellow), Reviewer:
Here, are you referring to “writing stamina“ perhaps? Or, if you really do mean “writing experience,“ then are you talking about years of experience or different types of genres? Can you please elaborate on this?

Caret, Reviewer:
a

Caret, Reviewer:
how

Strikeout (red), Reviewer:
in

Caret, Reviewer:
into

Highlight (yellow), Reviewer:
How is “posed cognitive load” different from “cognitive load” as you’ve been discussing it so far?

Strikeout (red), Reviewer:
, 2021

Caret, Reviewer:
(2021)

Caret, Reviewer:
one’s

Highlight (yellow), Reviewer:
Again, please be sure to define this earlier.

Strikeout (red), Reviewer:

Caret, Reviewer:
the

--- Page 10 ---

Strikeout (red), Reviewer:
,

Strikeout (red), Reviewer:
An

Caret, Reviewer:
the

Caret, Reviewer:
the

--- Page 11 ---

Highlight (yellow), Reviewer:
How many further to find these measures here? You’ve talked about both self reports and physiological measures of cognitive load, so you might want to summarize them briefly.

Highlight (yellow), Reviewer:
While I am not an L2 researcher, it would seem to me that someone who is interested in L2 would want to have even more attention put on the differences between different types of writing tasks, from the colloquial to the more formal. Also, it would seem to me that some question of a person’s current measure of language proficiency would be imperative as well.

Highlight (yellow), Reviewer:
On the “posed” cognitive load, or on the “actual” cognitive load? I think this is the first time you use this phrase, and I wonder if there is clarification that you need to offer earlier on?

Highlight (yellow), Reviewer:
Given that this is the conclusion that you draw, I wonder if you might be able to create some kind of diagram showing the intersection of these various factors and complexities?

Highlight (yellow), Reviewer:
While I certainly do not expect you to go into extensive detail about what these kinds of studies might look like, I would strongly suggest that you offer at least some hints about how researchers might methodologically design such studies for the future.

You really give this short shrift in the final paragraph, leaving your readers to try and figure out exactly what it is you are suggesting. I strongly encourage you to outline at least a few different ideas for qualitative and quantitative studies that would align with your model.

Caret, Reviewer:
a

Highlight (yellow), Reviewer:
It might also suggest that you need to consider sociocultural context as well as language proficiency.

Highlight (yellow), Reviewer:
It is not entirely clear what you mean here by giving it a cutting edge.

--- Page 12 ---

Strikeout (red), Reviewer:
syllabus

Caret, Reviewer:
instructional

Round 2

Reviewer 2 Report

Dear Colleagues,

Thank you for your attention to my earlier review, and the revisions you have made. Overall, I agree that your manuscript has improved and offer only a few minor suggestions as you prepare for publication: 

- In the table on page 4, I wouldn't necessarily agree that writing on a computer is a "monotonous experience"

- Line 525 needs to begin "An encouraging" 

- In Line 605, remove "empirically" from "empirically assess" and perhaps replace with "formatively as well as more formally assess"  (unless you mean that individual writers will be empirically assessed by someone other than a teacher). 

- Thanks for including Figure 1, "Writing modality as a task complexity factor." I would also encourage you to include voice-to-text dictation in there as one of the skills. You might provide a reference to Coiro as a way to indicate you are building something similar. 

Again, thank you for your revisions and I wish you the best of luck as you prepare your manuscript for publication.

Author Response

Dear Editors and Reviewers,

Thank you so much for your suggestions which have helped us improve our manuscript. Below we provide the table with the comments in which we explain how we attended to your suggestions.

With our best regards,

Authors

Reviewer comments

Solution

In the table on page 4, I wouldn't necessarily agree that writing on a computer is a "monotonous experience"

We eliminated “monotonous”

Line 525 needs to begin "An encouraging" 

DONE

In Line 605, remove "empirically" from "empirically assess" and perhaps replace with "formatively as well as more formally assess"  (unless you mean that individual writers will be empirically assessed by someone other than a teacher). 

Concerning this suggestion, we finally decided to leave the original wording “empirically assess”. With this expression we want to say that cognitive load and other relevant variables have to be measured by researchers in their studies/experiments.

Thanks for including Figure 1, "Writing modality as a task complexity factor." I would also encourage you to include voice-to-text dictation in there as one of the skills. You might provide a reference to Coiro as a way to indicate you are building something similar. 

We very much appreciate your suggestions concerning Figure 1.

As for the suggestion to include voice-to-text dictation as one of relevant skills, we finally decided not to include it, provided we could not find any empirical evidence or theoretical justification that voice-to-text dictation can have impact on writing performance in different writing modalities. For this reason, we decided not to include this skills in the Figure, but we will surely consider the potential relevance of voice-to-text in our future research.

Concerning the suggestion to provide a reference to Coiro, unfortunately we could not find a way to connect Coiro`s figure to our own Figure. The only common issue that our Figure 1 shares with Coiro`s figure is the form of a mind map, which is genèrica format. However, there is no conceptual connection between the two figures. For this reason, we decided not to provide reference to Coiro.